# Controlling the counterintuitive optical repulsive thrust of nano dimers with counter propagating type waves and background medium

Sudipta Biswas[1,2], M. R. C. Mahdy[2]*, Saikat Chandra Das[2,3], Md. Ariful Islam Bhuiyan[2], Mohammad Abir Talukder[2]

1 Department of Electrical and Computer Engineering, Baylor University, Waco, Texas, United States of America, 2 Department of Electrical & Computer Engineering, North South University, Bashundhara, Dhaka, Bangladesh, 3 Abbe School of Photonics, Friedrich Schiller University Jena, Jena, Germany

☯ These authors contributed equally to this work.
* mahdy.chowdhury@northsouth.edu

**Data Availability Statement:** All relevant data are within the paper and its Supporting Information files. Additional data will be available on request.

## Abstract

This work focuses on the utilization of counter-propagating plane waves for optical manipulation, which provides a unique approach to control the behavior of Rayleigh and Dipolar nanoparticles immersed in a homogeneous or heterogeneous medium. Our study presents an interesting finding of a repulsive force between plasmonic-chiral heterodimers where the particles move away from each other in both near and far field regions. Interestingly, this repulsive thrust supports the wave like nature of light for the case of homogeneous background but particle type nature of light for heterogenous background. At first, we have investigated the theory underlying the optical trapping of the chiral particle and the impact of this phenomenon on the overall repulsive behavior of the heterodimers placed in air (homogeneous) background. After that, our proposed set-up has further been investigated putting in air-water interface (heterogenous background) and by varying light angle only a little bit. Our observation for this interface case is suggesting the transfer of Minkowski momentum of photon to each optically pulled Rayleigh or dipolar particle of the dimer set, which ultimately causes a broad-band giant repulsive thrust of the dimers. However, in absence of the other particle in the cluster, a single half-immersed particle does not experience the pulling force for the broad-band spectrum. The 'common' reason of the observed repulsive thrust of the dimers for both the aforementioned cases has been attributed to "modified" longitudinal Optical Binding Force (OBF). Technically, this work may open a new way to control the repulsion and attraction between the nanoparticles both in near and far field regions by utilizing the background and the counter-propagating waves. We also believe that this work manifests a possible simple set-up, which will support to observe a background dependent wave 'or' particle nature of light experimentally.

**Funding:** M.R.C.Mahdy acknowledges the support of NSU CTRGC Grant 2022-23 (approved by the NSU authority).

**Competing interests:** The authors have declared that no competing interests exist.

# 1. Introduction

Particles can be transported or trapped through the transverse spatial gradient when subjected to a focused laser beam [1]. Because electromagnetic (EM) waves may transport and transmit momentum to the particles, this phenomenon occurs. A revolution in optical manipulation has been ignited by the idea of controlling particle motion with light, which has resulted in numerous studies and real-world applications involving various particle kinds, from cold atoms to living cells [2]. Numerous scientific disciplines have investigated optical manipulation in depth, with a focus on the Curl, Gradient, and Scattering forces along with Lorentz force distributions [1, 3–5]. The creation of multiple traps made by interfering waves [6] and spatial light modulators [7] to manipulate numerous items at once was facilitated by the discovery of optical tweezers [1].

In this article, we introduce a novel observation concerning light-matter interaction, focusing on controlling the counterintuitive repulsive force between plasmonic-chiral heterodimers in both near and far field regions when subjected to linearly polarized counter-propagating plane waves. We have explored the influence and modulation of this repulsive force on the dimers by investigating two different background types—homogeneous and heterogeneous—in our configurations.

The wave nature of light successfully explains several properties, especially in free space propagation, double slit experiment and so on. In contrast, the particle type nature of light, as proposed by A. Einstein in 1905 (for which he won the Nobel Prize in 1921) [8], is supported by experiments like the photoelectric effect [9] and Compton effect [10]. The complexity of wave-particle dual nature of light mainly arises during the different cases of light-matter interactions.

In our proposed set-up as expressed in Fig 1 involving heterodimers in a homogeneous medium with two counter-propagating light beams, the observed repulsive thrust cannot be solely attributed to the particle-like nature of light. Instead, the wave nature of light seems to play a vital role in this light-matter interaction.

On the other hand, in the forthcoming Fig 2(A), we depict a configuration where particles are placed at the air-water interface, representing a heterogeneous background. This setup was explored using two counter-propagating waves with a slight angular variation from both sides (15 degrees from the interface), revealing a broad-band repulsive thrust attributed to the individual pulling of each particle across the full spectrum. Here, "pulling force" refers to the force directing a particle towards its nearest unperturbed light source. Our results indicate the transfer of Minkowski momentum of photon [11–14] to each half-immersed Rayleigh or dipolar particle. Notably, this behavior, supporting the particle type momentum exchange, presents a notable exception in our proposed dimer setup in Fig 2(A) and cannot be solely attributed to the wave nature of light, aligning more with the photon picture of light [8–10]. However, it is really important to note for this case: in absence of other particle in the dimer set of our proposed set-up, a single half-immersed Rayleigh particle does not experience the pulling force for the broad-band spectrum. This last fact requires a more in-depth reasoning along with the aforementioned explanation of the transfer of Minkowski momentum of photon.

As a result, considering and analyzing all the aforementioned results, the 'common' reason of the observed repulsive thrust of the dimers for both the aforementioned cases (homogeneous background and heterogeneous background) have been attributed to "modified" longitudinal Optical Binding Force (OBF).

OBF, a phenomenon discovered in 1989, involves light scattering by micro-objects interacting with light's mechanical properties, resulting in the spatial distribution of separated nano-sized objects [15–19]. Despite its potential in optical manipulation, OBF[15, 20–24] has been

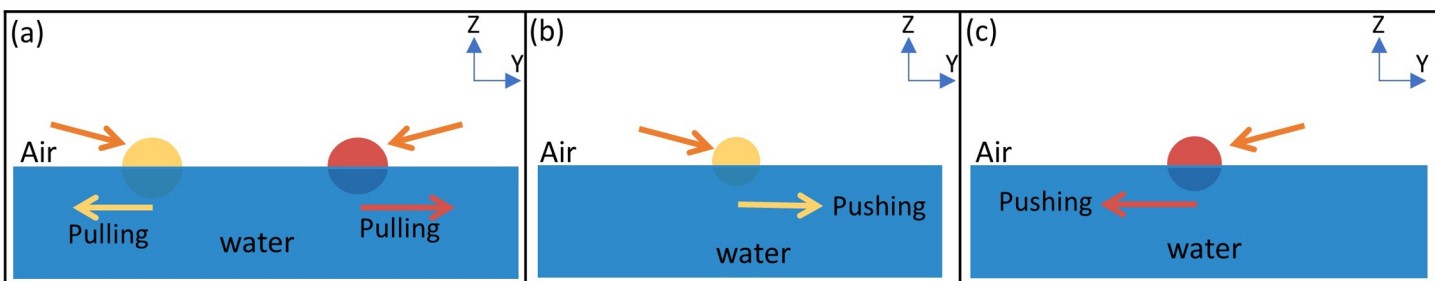

**Fig 1.** (a-c) depict plasmonic-chiral dimers illuminated by linearly polarized plane wave(s) in various configurations, illustrating the direction of wave propagation and the positioning of the plasmonic and chiral particles. Fig 1(d) and 1(e) specifically show the particles under illumination from counter-propagating plane waves, highlighting different responses in particle movement. In all cases, the left sphere is plasmonic and the right sphere is chiral.

**Fig 2.** (a) Plasmonic (yellow) and chiral (red) objects half immerged in air-water interface and the set up is illuminated by a X-polarized counter propagating type waves with incident angle 15 deg. Each Rayleigh particle in the set has experienced optical pulling force (towards the nearest unperturbed light source) and a broad-band repulsive binding force has been observed. (b) Single plasmonic object is half immerged in air-water and illuminated by X-polarized plane wave propagating along +Y axis with incident angle 15 deg. No pulling force has been observed. (c) Single chiral object is half immerged in air-water and illuminated by X-polarized plane wave propagating along -Y axis with incident angle 15 deg. No pulling force has been observed.

less explored. It's classified into near-field [25, 26] and far-field OBF [27, 28]. Yano et al. demonstrated control of lateral OBF between homodimer nanoparticles [22], and the impact of single electromagnetic and acoustic waves on conducting particle pairs was studied [29–32]. Our prior work showcased longitudinal OBF between plasmonic-chiral heterodimer with added plasmonic substrate and single incident light [33]. The behavior of a plasmonic-chiral dimer varies significantly depending on the presence of single or counter-propagating waves, with previous studies focusing on dielectric dimers and Bessel beams [18, 23, 34]. Distinctly, our work investigates 'modified' longitudinal OBF due to counter-propagating linearly polarized plane waves on a plasmonic-chiral heterodimer in both homogeneous and heterogeneous backgrounds.

In this study, we placed a plasmonic and chiral nanoparticle adjacently, immersed in either a homogeneous or heterogeneous background, and varied the interparticle distance in both near and far field regions. Upon illumination with counter-propagating plane waves, the particles exhibited counterintuitive optical thrust, moving apart in both types of media, with notable broad-band repulsion at the interface [cf. Figs 1 and 2]. Our findings suggest that such behavior may not be replicable with arbitrarily shaped homodimers and heterodimers in the presence of two counter-propagating waves.

This investigation addresses a crucial advantage of implementing counter-propagating plane waves for controlling longitudinal OBF among dimers (immersed in homogeneous and also heterogeneous medium). Interestingly, this counterintuitive repulsive thrust at the same time supports both the wave-like nature (i.e., for homogeneous background) and particle-type momentum exchange (i.e., for heterogeneous background) nature of light. Technically, this work may open a new way to control the repulsion and attraction both in near and far field regions between the nanoparticles by utilizing the background and the counter-propagating waves. We also believe that this work manifests a possible simple set-up, which will support to observe a background dependent wave 'or' particle nature of light experimentally.

## 2. Optical setup

This article demonstrates a method to regulate repulsion in near and far-field inter-particle forces for both Rayleigh and Dipolar particles using two counter-propagating plane waves. Our investigation suggests that such behavior might not be replicable with arbitrarily shaped homodimers and heterodimers. In our setup, the chiral and plasmonic heterodimers are illuminated with a linearly polarized plane wave from both sides. Chiral objects are essential part of biological world [12, 35]; while, plasmonics are an effective tool effective in optical manipulation due to their induced charges [36–38]. Our proposed configuration provides a relatively simple optical setup that suggests a possible method for controlling the repulsion of two particles in both the near- and far-field regions.

The schematic diagrams and visual representations of our setups are provided in Fig 1 (background is air) and Fig 2 (background is air-water interface). We simulated configurations with plasmonic-chiral heterodimers of Rayleigh (r = 100nm) and Dipolar (r = 200nm) sizes. Fig 1A–1C illustrates the schematic, with Fig 1A and 1B depicting single linearly polarized plane waves in the "-y" or "+y" directions, and Fig 1(C) showing the dimer configuration illuminated by two plane waves in opposing directions. The polarization of both lights is maintained in the "+x" or "+z" direction. We varied the wavelength ($\lambda$) of the incident wave from 400 nm to 1000 nm and set interparticle distances at 200nm (near-field) and 700nm (far-field). For heterogeneous backgrounds (cf. Fig 2), the light angle was slightly adjusted (15 degrees from the interface).

The permittivity of gold, including real and imaginary components, was referenced from standard Palik data. The chiral nanoparticle had a refractive index of 1.45, with a chirality parameter set at $\kappa = +1$. The tests were simplified by utilizing small-radius particles, and the simulation software was not overtaxed because the resonance of the particles was not very powerful. The optical force was determined using numerical methods such as the coupled dipole approach [18] and finite element method through COMSOL Multiphysics [24].

## 3. Results and discussion

This paper aims to investigate the repulsive thrust of Plasmonic-Chiral heterodimer particles in presence of the continuous pushing force created by two counter propagating plane waves. In our configuration, as depicted in Figs 1A–1C and 3, the plasmonic nanoparticle is at the left, while the chiral nanoparticle is to the right. We used COMSOL Multiphysics to calculate the optical forces external to the volume of these nanoparticles, termed as "exterior" or "outside" optical forces. The 'time-averaged optical force' is calculated using the time-averaged Minkowski Stress Tensor [12, 25, 35, 39–43] at r = a+, considering the background fields of the scatterer with radius 'a'.

$$< \mathbf{F}_{\text{Total}}^{\text{Out}} > = \oint \left\langle \overline{\overline{\mathbf{T}}}^{\text{out}} \right\rangle . \, d\mathbf{s} \tag{1}$$

$$< \overline{\overline{\mathbf{T}}}^{\text{out}} > = \frac{1}{2} Re \left[ \mathbf{D}_{\text{out}} \, \mathbf{E}_{\text{out}}^* + \mathbf{B}_{\text{out}} \, \mathbf{H}_{\text{out}}^* - \frac{1}{2} \overline{\overline{\mathbf{I}}} \left( \mathbf{E}_{\text{out}}^* \cdot \mathbf{D}_{\text{out}} + \mathbf{H}_{\text{out}}^* \cdot \mathbf{B}_{\text{out}} \right) \right] \tag{2}$$

In Eq (1), the term "out" refers to the total external field of the scatterer, which is the combination of the incident field and the scattered field. The symbols "E", "D", "H" and "B" represent electric field, displacement vector, magnetic field and induced magnetic field vector, respectively. The symbol "$< >$" represents the mean time and $\overline{\overline{\mathbf{I}}}$ represents the unit tensor.

For plasmonic-chiral heterodimers, the longitudinal OBF is defined as $F_{\text{bind}(y)} = (F_{1(y)} - F_{2(y)})$, where positive values in OBF indicate attractive forces, but negative values indicate repulsive forces. The subscripts (y), (1), and (2) indicate the "y" component of the OBF, the left and right spheres, respectively. In this paper, these forces represent the OBF in a forward (F-F) configuration. For the chiral nanoparticles used in this paper, the basic constitutive relationship is given by the following Eqs [2, 35, 43, 44]:

$$\begin{bmatrix} \mathbf{D} \\ \mathbf{B} \end{bmatrix} = \begin{bmatrix} \varepsilon_r \varepsilon_0 & i\kappa/c \\ i\kappa/c & \mu_r \mu_0 \end{bmatrix} \begin{bmatrix} \mathbf{E} \\ \mathbf{H} \end{bmatrix} \tag{3}$$

Here in Eq (3), $\varepsilon_{(r)}$ and $\mu_r$ represent the relative permittivity and permeability of the chiral material, respectively.

The chirality parameter is denoted by $\kappa$, which is controlled by the inequality $\kappa^2 < \varepsilon_r \mu_r$ [29]. Furthermore, $\varepsilon_0$, $\mu_0$ and c represents the permittivity, permeability, and speed of light in vacuum. The real and imaginary parts of the permittivity of gold have been taken from the standard Palik data for the plasmonic object.

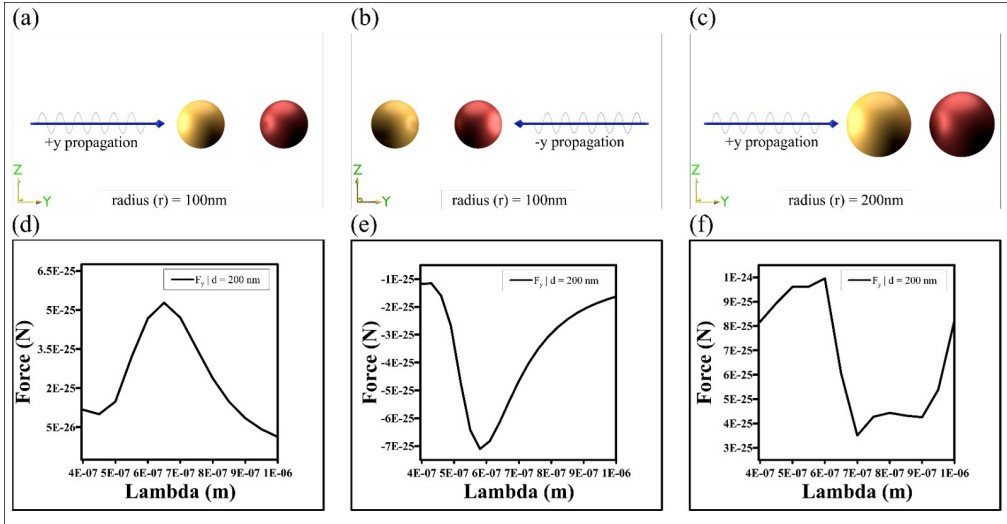

**Fig 3.** a) "x" polarized plane wave is propagating in the "+y" direction towards the nanoparticles [r = 100nm]. b) similar configuration as Fig 3(a), but the light propagation direction is towards "-y" direction. c) Similar light propagation direction as Fig 3(a), but the particles' radius is r = 200 nm. (d-f) shows the longitudinal OBF for the setups of Fig 3a-3c, respectively. (In all cases, left sphere and right spheres are plasmonic and chiral respectively).

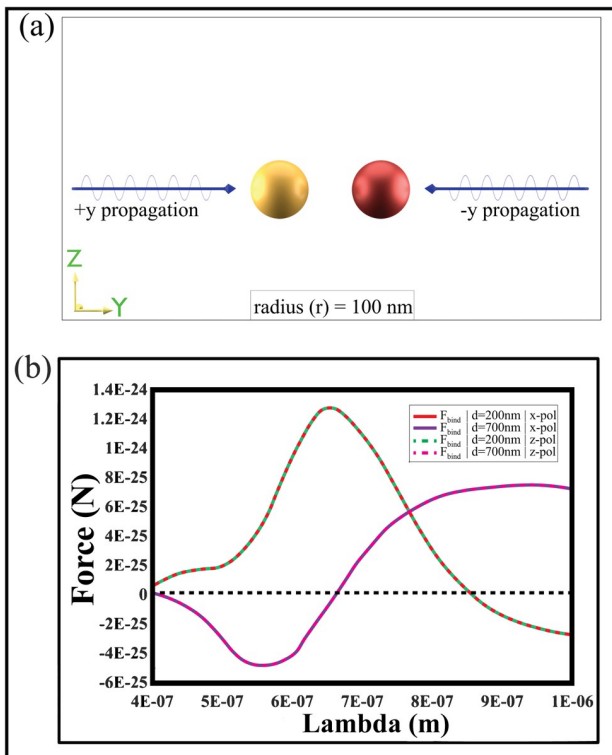

**Fig 4.** (a) "x" or "z" polarized plane waves are propagating towards both "+y" and "-y" directions hitting the plasmonic(left) and chiral(right) nanoparticles from both sides. The particles lie in the Rayleigh region, and their radius is r = 100nm. (b) The OBFs (along the "y"—axis) are shown between the two-hybrid dimers for Fig 4(a). The gap distance between the dimers is 200 nm and 700 nm by varying wavelengths.

### 3.1 Plasmonic chiral hybrid heterodimers placed in air (homogeneous) background in presence of counter propagating waves: Face to face repulsive force for particles in rayleigh range

Before delving into our primary setup: plasmonic-chiral heterodimers in presence of oppositely polarized plane waves, we will address several relevant configurations that will aid in understanding the internal physical mechanism of our proposed repulsive thrust of particles.

Firstly, considering the heterodimer setup of plasmonic-chiral particles with a single linearly polarized plane wave (cf. Fig 3), no repulsive force is observed between the plasmonic and chiral heterodimers for particles in both Rayleigh and Dipolar regions using a single light source. In this scenario [cf. S1 Section in S1 File for more details], particles are illuminated from one side, experiencing an optical pushing force due to the momentum carried by photons (the quanta of light) [45–47]. The force between these particles in presence of a single wave from a particular side is termed as longitudinal OBF.

Finally, considering the plasmonic-chiral heterodimer configuration in the presence of linearly polarized counter-propagating plane waves, we observe both attractive and repulsive forces in our final configuration. In spite of the fact that several properties of light can be clarified effectively considering it as electromagnetic wave (classical wave like picture) for free space propagation, the light-matter interaction picture emphatically underpins both the wave and particle nature of light. As far as we will get it, the observed repulsive thrust of the heterodimers in presence of two counter engendering light beam cannot be exclusively credited to photon or particle like picture of light. [8–10]. In differentiate, it shows that the wave nature of light plays a crucial part for the light-matter interaction of our proposed set-up. The observed repulsive force between the plasmonic-chiral heterodimer is quite counter-intuitive and the reason can be attributed to "modified" longitudinal OBF. The term "modified" is used due to the presence of two oppositely propagating waves, contrasting with a single propagating wave scenario, with additional reasons detailed in Section 4. The schematic diagram of our proposed layouts for particles in the Rayleigh region is shown in Fig 4(A), depicting lights propagating towards +y and -y, with consistent polarization in either +x or +z direction and wavelength ($\lambda$) ranging from 400 nm to 1000 nm.

In Fig 4(B), the modified OBF achieved with the new arrangement is displayed for Rayleigh particles (r = 100nm) at interparticle distances of 200nm and 700nm. A reversal in the modified longitudinal OBF is observed for both distances upon introducing a second plane wave source. Specifically, at an interparticle distance of 200nm (for both x and z polarizations of light), the OBF is initially negative (repulsive), peaking at $\lambda$ = 550nm. However, it changes polarity at $\lambda$ = 650nm, becoming positive (attractive), with the attractive force peaking at $\lambda$ = 1000nm. A similar reversal in the OBF polarity is also observed for an interparticle distance of 700nm. Now the question is why we get such behavior in presence of two counter propagating waves? In the following subsection, we will address the answer briefly.

**3.1.1 Underlying physical mechanism behind the altered longitudinal OBF.** In this case, we have two sources of linearly polarized plane waves. Thus, both the plasmonic and chiral particles are being pushed equally by the light from both sides. Despite this, a repulsive behavior is observed. Examining the electric field profiles in Fig 5 reveals an abrupt change in the electric field when the OBF reverses [cf. force values in Fig 4(B)]. At a lower wavelength with interparticle distance d = 200nm, where the OBF is repulsive, both positive and negative charges accumulate on the plasmonic and chiral nanoparticles, resulting in an overall repulsive force between them [Fig 5(A)]. Contrastingly, Fig 5(B) illustrates that at higher wavelengths, where OBF is attractive, the induced electric field exhibits attractive behavior due to the accumulation of opposite charges on the nanoparticles. The electric field profiles for far-field

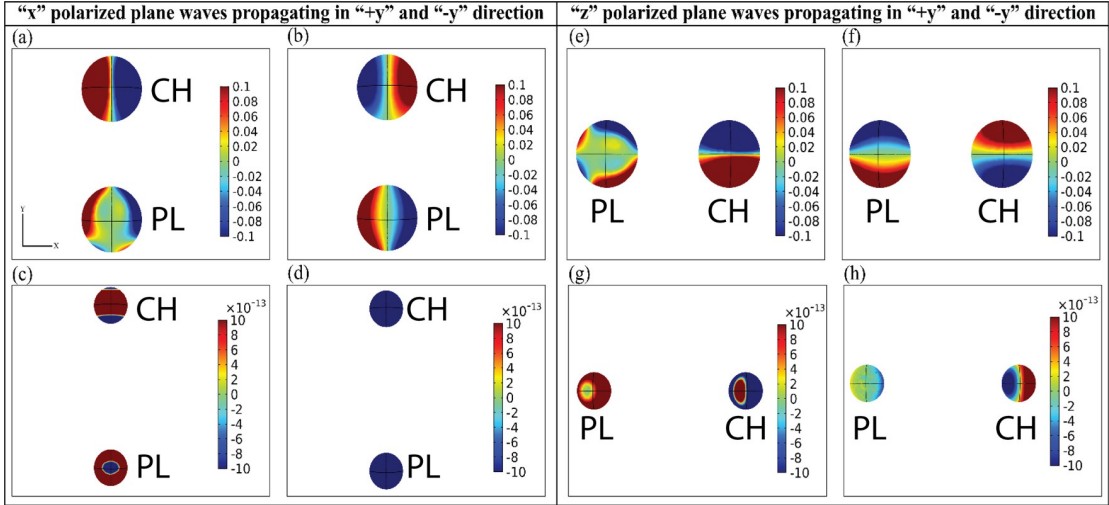

**Fig 5.** (a-d) electric field profiles for configuration of Fig 4(A) when "x" polarized plane waves are propagating in the "+y" and "-y" direction. (a) interparticle distance, d = 200 nm at wavelength 450nm (b) interparticle distance, d = 200 nm at wavelength 850nm (c) interparticle distance, d = 700 nm at wavelength 450nm (d) interparticle distance, d = 700 nm at wavelength 900nm. (e-h) electric field profiles for configuration of Fig 4(A) when "z" polarized plane waves are propagating in "+y" and "-y" direction. (e) interparticle distance, d = 200 nm at wavelength 450nm (f) interparticle distance, d = 200 nm at wavelength 850nm (g) interparticle distance, d = 700 nm at wavelength 450nm (h) interparticle distance, d = 700 nm at wavelength 900nm. Please note that 'PL' & 'CH' indicates plasmonic and chiral object respectively.

regions of this configuration are depicted in Fig 5C–5H displays the profiles when the lights are polarized towards the +z direction for interparticle distances of 200nm and 700nm, respectively.

When illuminated by a standing wave, both plasmonic and chiral spheres experience an increase in local field enhancement due to emitted photons. Surface plasmons are formed as electromagnetic waves propagate through a plasmonic object, generating a plasmonic force.

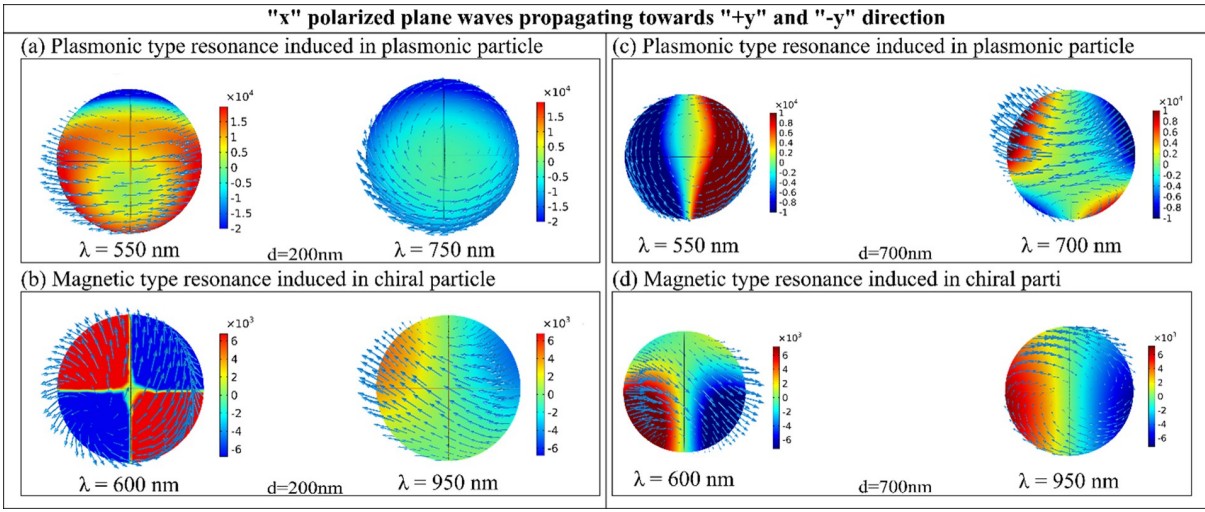

**Fig 6.** (a-d) induced current density profiles on the plasmonic and chiral nanoparticles when "x" polarized plane waves are propagating towards "+y" and "-y" direction [cf. Fig 4(A)] (a) for interparticle gap distance, d = 200nm at wavelength 550nm and 750nm (b) for interparticle gap distance, d = 200 nm at wavelength 600nm and 770nm (c) for interparticle gap distance, d = 700 nm at wavelength 550nm and 700nm (d) for interparticle gap distance, d = 700 nm at wavelength 600nm and 950nm.

This force originates from charges induced by the surface force, a result of the presence of free electrons on the nanoparticle surface [35, 48]. The Surface Plasmon Resonance (SPR) of plasmonic nanospheres is influenced by factors such as size, shape, composition [49], and arrangement within nanoparticle assemblies [50]. A radius of 100nm was chosen for both plasmonic and chiral hybrid dimers, ensuring that all electrons within the nanostructure could be rapidly excited, contributing to the plasmon's oscillation [51]. This leads to a stronger interaction between Plasmonic nanoparticles and the Electromagnetic (EM) wave [16, 52]. Additionally, the presence of two light sources induces electron oscillation between the two heterodimers, resulting in significant charge flow. This enhances the electric field strength, creates stronger coupling and resonance, which in turn leads to intense multiple scattering and a reversal of OBF.

We have identified an additional phenomenon contributing to the reversal of longitudinal OBF. Fig 6A–6D illustrates the current density profile for plasmonic and chiral particles, correlating with the configuration in Fig 4(A), where both lights are polarized towards the x-direction. The arrows in Fig 6A–6D represent the direction of induced electric current flow. Examining Fig 6A and 6B, we notice variations in current density for both plasmonic and chiral particles at different wavelengths, with an interparticle distance, d = 200 nm. When the OBF is repulsive, a rotating current density pattern is noticeable on the chiral particle, indicative of an induced magnetic type resonance [cf. Fig 6(B)]. Such a phenomenon is absent in the plasmonic particle at these wavelengths. Conversely, in regions where the OBF is attractive, the chiral particle's rotating pattern vanishes, and a rotating pattern emerges on the plasmonic particle, indicating plasmonic type resonance [cf. Fig 6(A)]. Similarly, with an interparticle distance, d = 700 nm, a comparable rotating pattern is observed on the particles at various wavelengths, as depicted in Fig 6C and 6D. A similar rotating pattern is also observed when both the propagating lights are polarized towards the z-direction, as depicted in Fig 7A–7D.

Furthermore, we have extended our investigation on the chiral object in our dimer set-up. Our arrangement's $F_y$ force of the chiral object makes the phenomenon more interesting. The $F_y$ force for the chiral object has been shown in S2A and S2B Fig (S1 File). We observe that the $F_y$ force of the chiral nanoparticle becomes negative from positive exactly at the wavelength where the overall binding force is reversed. We refer to this specific state as a trapping position, which is analytically detailed in the subsequent section.

**3.1.2 Analytical discussion behind the trapping force on the chiral particle (placed in air) in presence of counter propagating waves: Its impact on overall binding force.** Before explaining the concept of trapping force, we first studied the behavior of chirality in an interfering wave, with the analytical derivation provided in S3 Section (S1 File). In this section, however, our focus will be on analytically discussing the trapping force experienced by the chiral object. Here, $p$ and $m$ represent the induced electric and magnetic moments for a chiral nanoparticle, as detailed below:

$$p = \alpha_e E + \chi H$$

$$m = -\chi E + \alpha_m H \tag{4}$$

The time average optical force on a chiral nanoparticle in free space can also be written as using the following equation.

$$\langle F \rangle = \frac{1}{2} \, Re\left[ p(\nabla \otimes E^*) + m(\nabla \otimes H^*) \, - \, \frac{ck^4}{6\pi}(p \times m^*) \right] \tag{5}$$

Here, This equation includes the incident electric field vector (E), the incident magnetic

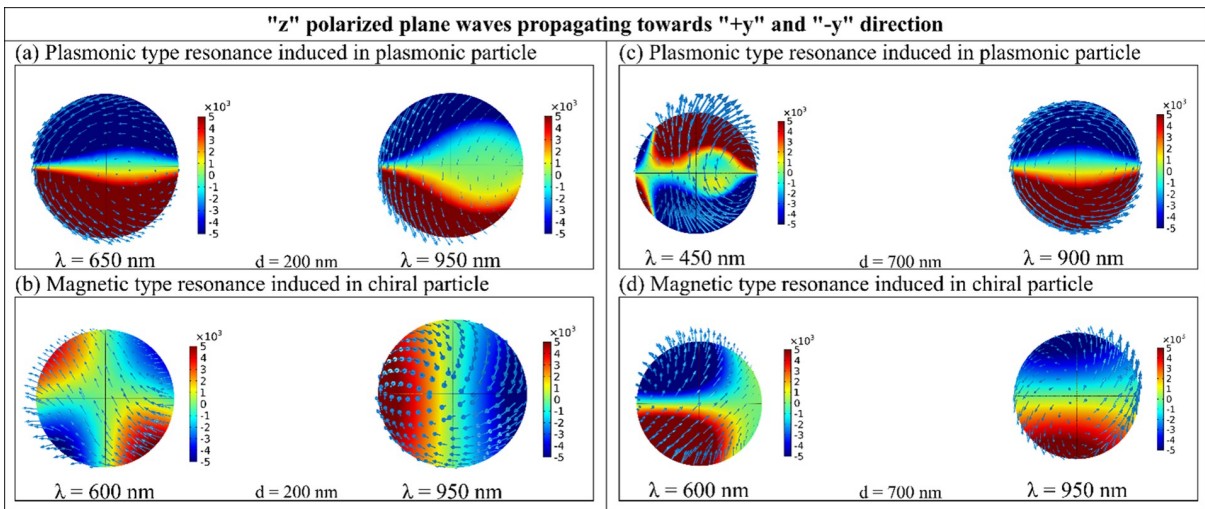

**Fig 7.** (a-d) induced current density profiles on the plasmonic and chiral nanoparticles when "z" polarized plane waves are propagating towards "+y" and "-y" direction[cf. Fig 4(a)] (a) for interparticle gap distance, d = 200nm plasmonic particle at wavelength 650nm and 950nm (b) for interparticle gap distance, d = 200 nm chiral particle at wavelength 600nm and 770nm (c) for interparticle gap distance, d = 700nm plasmonic particle at wavelength 450nm and 700nm (d) for interparticle gap distance, d = 700nm chiral particle at wavelength 600nm and 950nm.

field vector (H), and the wave number (k). The symbol "⊗" represents a binary product. The equation consists of three terms corresponding to contributions from electric dipoles, magnetic dipoles, and direct interactions between electric dipoles. We separate the force into the dipole part $\mathbf{F_{EM}}$ (first two term of the Eq 5) and interaction part $\mathbf{F_{INT}}$ (last term of the Eq 5). From equation (S3) and (S4) in S1 File we get the expression for $\langle \mathbf{F_{EM}} \rangle$ ***and*** $\langle \mathbf{F_{INT}} \rangle$.

$$\langle \mathbf{F_{EM}} \rangle = \frac{1}{2} Re[\alpha_e] \nabla |\mathbf{E}|^2 + Re[\alpha_m] \nabla |\mathbf{H}|^2 + 2\omega(\mu_0 Im[\alpha_e] + \varepsilon_0 Im[\alpha_m])\overline{\mathbf{p}} +$$
$$\frac{c^2}{\omega} Re[\chi] \nabla \overline{K} - 2Im[\chi]\overline{\Phi} + \frac{2}{\omega\varepsilon_0} Im[\alpha_e] \nabla \times \overline{\Phi_E} + \frac{2}{\omega\mu_0} Im[\alpha_m] \nabla \times \overline{\Phi_M} -$$
$$Im[\chi] \nabla \times \overline{\mathbf{p}}$$

(6)

$$\langle \mathbf{F_{INT}} \rangle = -\frac{ck^4}{6\pi} \left( 2\left( Re[\alpha_e \alpha_m^*] + |\chi|^2 \right)\overline{\mathbf{p}} - Im[\alpha_e \alpha_m^*] Im[\mathbf{E} \times \mathbf{H}^*] \right) -$$
$$\frac{2ck^4}{3\pi\omega} \left( \frac{1}{\varepsilon_0} Re[\alpha_e \chi^*]\overline{\Phi_e} + \frac{1}{\mu_0} Re[\alpha_m \chi^*]\overline{\Phi_m} \right)$$

(7)

$\overline{\mathbf{p}} = \frac{1}{2} Re[\mathbf{E} \times \mathbf{H}^*]$ is the time average poynting vector; $\overline{\Phi_E} = -\frac{\omega}{4} Im[\mathbf{E} \times \mathbf{E}^*]$ and $\overline{\Phi_m} = -\frac{\omega}{4} Im[\mathbf{H} \times \mathbf{H}^*]$ represent electric and magnetic parts of the chirality flow density.

The term in Eq (7) is this paper's core and leads to the chiral trapping force. However, the above equation is distinct from $\langle \mathbf{F_{EM}} \rangle$. In this case, for coupling to occur, the electric part $\Phi_e$ and magnetic part $\Phi_m$ are transferred with unlike efficiencies proportionate to $Re[\alpha_e \chi^*]$ and $Re[\alpha_m \chi^*]$ respectively.

If we replace the incident field Eqs (S4) in S1 File into (6) and (7). We get $\langle \mathbf{F_{EM}} \rangle = \mathbf{0}$ as $\nabla |\mathbf{E}|^2 + \nabla |\mathbf{H}|^2, \overline{\mathbf{p}}, \overline{K}$ and $\overline{\Phi}$ are all zero. Therefore, Total force of the chiral object is as

following Eq (8):

$$\boldsymbol{F} = \boldsymbol{F_{INT}} = -\frac{2ck^4}{3\pi\omega}\left(\frac{1}{\varepsilon_0}Re[\alpha_e\chi^*]\overline{\Phi_e} + \frac{1}{\mu_0}Re[\alpha_m\chi^*]\overline{\Phi_m}\right) = \frac{E_{inc}^2 ck_o{}^4}{3\pi}\left(-Re[\alpha_e\chi^*] + \frac{\varepsilon_0}{\mu_0}Re[\alpha_m\chi^*]\right)sin(2k_0 y)\hat{\boldsymbol{y}} \quad (8)$$

For any standard dual symmetric particle this force will be zero as the components of the electric and magnetic force cancel each other. Thus, we get a trapping force for chiral object.

The trapping type force of the chiral object plays a crucial role in inducing repulsion within the overall binding force. As illustrated in S2 Fig (S1 File), we have depicted the relationship between the binding force between two particles (placed in air) and the trapping type force of the chiral object. The phenomenon of optical binding can be influenced by the movement of either one or both of the particles. In our specific case, the chiral particle on the right is nearly trapped. As a result, the nature of the binding force, whether attractive or repulsive, is contingent upon the movement of the plasmonic particle. This occurrence has a substantial impact on the overall binding force. Remarkably, even when the chiral object is trapped at wavelength ranges around 650nm for near field and 890nm for far field (refer to S2 Fig in S1 File), we still witness an increase in the overall OBF. It is precisely at this point that the overall binding force undergoes a reversal.

Therefore, with the reversal of OBF, induced magnetic type and plasmonic type resonance have been observed on the chiral and plasmonic particles. This abrupt shift of current density, the effect of electric charges, and the trapping of the chiral nanoparticle are mainly responsible for the counterintuitive behavior of the particles. Thus, we achieved a 'direct' reversal of OBF between the plasmonic and chiral particles (placed in air) in the Rayleigh region. An in-depth analysis regarding the reversal of OBF within the dipolar region can be found in S4 Section of the S1 File.

## 4. Plasmonic chiral hybrid heterodimers at the air-water interface (heterogeneous medium) in presence of counter propagating waves: Face to face broadband repulsive force along with individual pulling force

When the chiral-plasmonic heterodimer is placed in a homogeneous background medium, it is possible to observe the reversal of longitudinal OBF in presence of two counter propagating waves. However, it is not possible to achieve the repulsion between the two particles for a broad-band region. In this section, we are going to propose a technique to achieve the broadband repulsive force for plasmonic-chiral nanoparticles by utilizing the background and two counter-propagating waves (but slightly varying the angle of light [15 degree from the interface] for both the sources).

In a homogeneous background medium, observing the reversal of longitudinal OBF is possible when a chiral-plasmonic heterodimer is present amidst two counter-propagating waves. However, achieving repulsion between the two particles across a broad-band region remains unattainable. In this section, we propose a method for attaining a broad-band repulsive force between plasmonic-chiral nanoparticles by employing a background and two counter-propagating waves, with a slight variation in the angle of light (15 degrees from the interface) for both sources. We have observed that when the plasmonic-chiral heterodimer set-up is placed at the heterogenous background like air-water interface (as shown in Fig 2(A)), each Rayleigh particle in the set-up experiences optical pulling force. *Here pulling force means the force on a particle towards the unperturbed nearest light source of that particle*. Ultimately, this leads to a broad-band giant repulsive OBF of the nano-dimers (cf. Fig 8(A) [the near field case]). However, obtaining optical pulling force for a half-immersed Rayleigh particle is quite

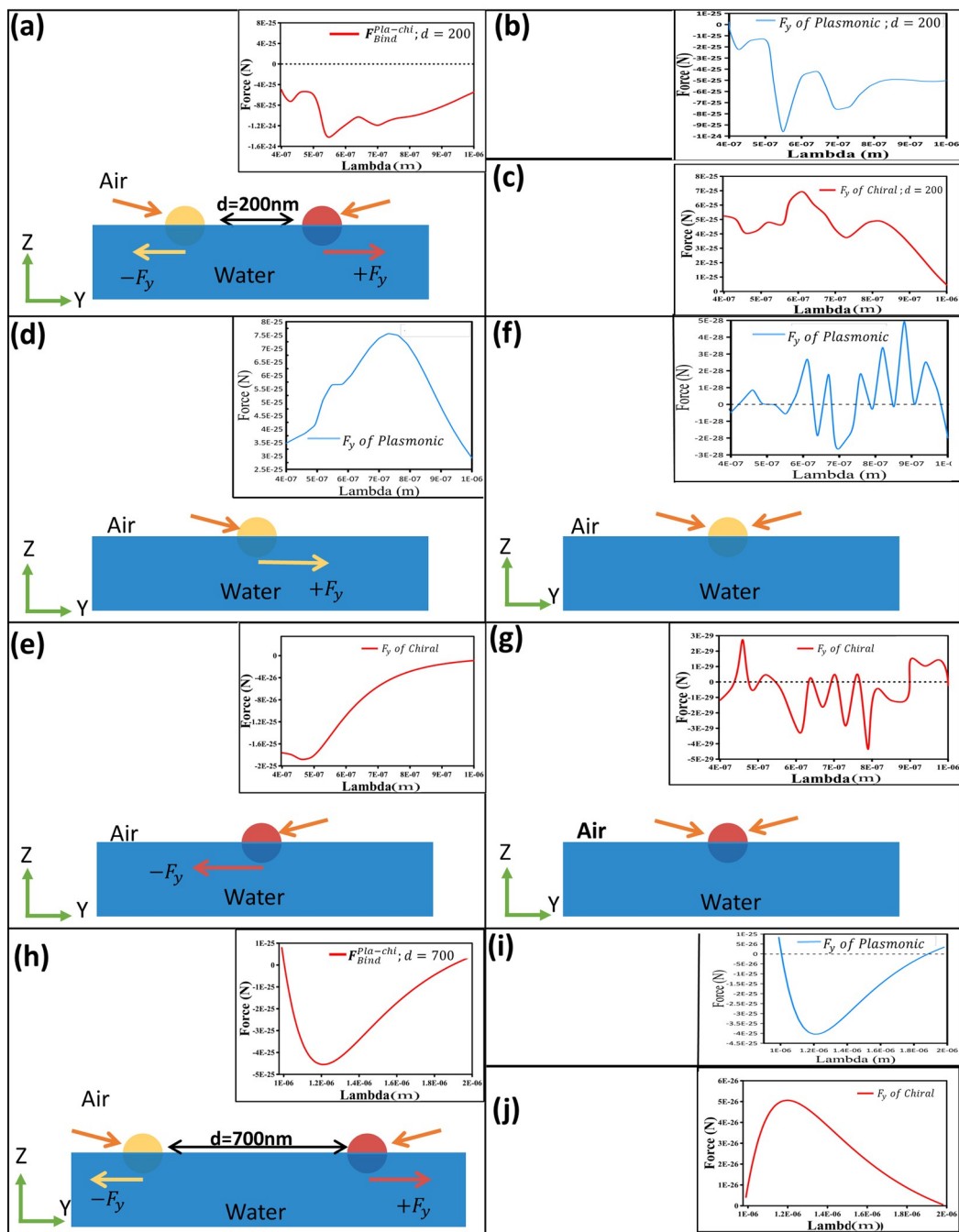

**Fig 8.** (a) Near field OBF between plasmonic (gold) & chiral (red) objects immersed in half air-water and the setup is illuminated by x polarized counter propagating type waves (15-degree angle from the air-water interface). Distance between the particles is set to 200nm (b) $F_y$ force on the plasmonic object from the dimer setup of (a). (c) $F_y$ force on chiral object from the dimer setup of (a). (d) $F_y$ force acting on a single half immersed plasmonic object for single light source (pushing force) (e) $F_y$ force acting on a single half immersed chiral object for single light source (pushing force) (f) $F_y$ force acting on a single half immersed plasmonic object with two light sources. This setup is illuminated by x polarized counter propagating type waves (15-degree angle from the air-water interface). (g) $F_y$ force acting on a single half-immersed chiral object with two light sources. This setup is illuminated by x polarized counter propagating type waves (15-degree angle from the air-water interface). (h) Far field OBF between plasmonic (gold colored) & chiral (red colored) objects immersed in half air-water and the setup is illuminated by x polarized counter propagating type waves (15-degree angle from the air-water interface). Distance between the particles is set to 700nm (i) $F_y$ force on plasmonic object from the setup of (h). (j) $F_y$ force on chiral object from the setup of (h).

counterintuitive [12] because to the best of our knowledge, interfacial tractor beam effect works only for Mie or bigger sized objects [11–13]. This counter-intuitive phenomenon has led to the broad-band repulsive thrust (for both the near field and far field regions); and we have tried to explain the possible physical reasons next.

Let us first consider the near field case shown in Fig 8(A)–8(G). It is important to note, while the repulsive behavior of the plasmonic-chiral dimer setup in a homogeneous medium can be explained through the wave picture of light, addressing the pulling of individual Rayleigh particles in a heterogeneous medium necessitates the introduction of the photon picture. In an air-water interface, a Mie-sized dielectric particle undergoes optical pulling force owing to the linear increase of photon momentum at the particle-water interface [11–13]. This linear increase in photon momentum aligns with the proposal of Minkowski, and its experimental validation for dielectric particles stands as significant evidence supporting the transfer of Minkowski momentum of photons [11–14], contrasting with Max Abraham's alternate theory implying the decrease of linear momentum of photons in a material medium [53, 54]. However, the interfacial tractor beam concept does not apply to a half-immersed plasmonic Mie particle [13], attributed to the inherent loss or absorption of the plasmonic particle [13]. Yet, this discrepancy does not invalidate the idea of transferring Minkowski photon momentum, as detailed in [13]. For instance, increasing the refractive index of the hypothetical second medium [13] could enable the observation of pulling in a Mie-sized absorbing half-immersed particle. However, to the best of our knowledge, no work has shown the validity of interfacial tractor beam concept for the Rayleigh particles. Even a half immersed dielectric non-absorbing Rayleigh particle cannot be pulled. As a result, controlling the repulsion of more complicated half-immersed Rayleigh particles (such as plasmonic particle and chiral particle) seems quite counter-intuitive.

Fig 8(D) and 8(E) illustrate that such repulsion does not occur for independently placed half-immersed Rayleigh-sized single plasmonic or chiral particles. Nevertheless, in our overall setup (as depicted in Fig 8(A)–8(C)), each Rayleigh particle experiences a broad-band optical pulling force towards the nearest unperturbed light source. Increasing the refractive index of the lower medium can indeed enhance the repulsive thrust for the setup shown in Fig 8(A).

In Fig 8(F) and 8(G), we analyze the time-averaged force on a single Rayleigh-sized plasmonic or chiral particle at the air-water interface, illuminated by two counter-propagating waves at a 15-degree angle from the interface. It is observed: while a single particle still does not experience optical pulling force, optical trapping type force (the value of force is extremely smaller than the single beam pushing force case or the case of OBF) arises naturally in such a set-up. It's crucial to note that the cone angle between the two linearly polarized beams is 150 degrees or above. This type of highly non-paraxial dual beam set-up can create the pulling (tractor beam) effect for the Mie sized particles [55, 56] (due to the increase of directional momentum of the emitted photon [55, 56]) and this has been verified experimentally in [57]. A detailed analysis regarding the mutual interaction term of dipolar force law is given in [58]. Interestingly, the setup in Fig 8(F) supports optical trapping type force even for highly absorbing Rayleigh particles, such as plasmonic ones. This trapping type force is observed to be effective across a broad-band region for a single particle in the setup [Fig 8(F) and 8(G)]. It (trapping type force) will be the key to creating mutual repulsion when another particle is introduced in such a set-up to create a perturbation (the binding force effect), which will be considered next.

Returning to our main setup in Fig 8(A), we observe from Fig 8(B) and 8(C) that each Rayleigh particle experiences optical pulling force (towards the nearest unperturbed light source) across a broad-band region, ultimately leading to the broad-band repulsive binding force of the dimers. This observation prompts a question: how does the trapping type force region for each particle, depicted in Fig 8(F) and 8(G), transform into a pulling force region? The answer

lies in the concept of OBF, which is indeed a different kind of force than the pure gradient or scattering force of light. Certainly, the presence of the other particle in the dimer set is impacting the dynamics of another particle. The generic approach to explain such dynamics based on well-known dipolar force model [18] for the 'simple' dipoles:

In this model, the incident field induces a dipole in both the left-side and right-side particles, with the right-side dipole oscillating with a phase shift kd (where d represents the interparticle distance) relative to the left-side one. The right-sided dipole then radiates backward toward the left sided dipole. This backward-scattered light reaches the oscillating left sided dipole with the total phase shift of 2kd. The force on the left sided particle arises from the interaction of the left sided dipole with the interference effect of the backward-scattered field and incident field. On the other hand, the force on the right sided particle pushes the right sided dipole along the wave propagation direction. In this way, the interaction between both 'simple' dipoles is usually repulsive in the very closely placed near field region.

In our configuration outlined in Fig 8(A), the particles involved are not mere 'simple' dipoles. Factors such as material response—including chirality, absorption of the plasmonic particle, magnetic response of the chiral particle, and the influence of the background medium due to the angle of light—all contribute significantly. A closer examination of each of the particle suggests that dipolar approximation (which was considered in [18, 23]) may not fully remain valid in higher wavelength regions of repulsive thrust (though it may remain valid in lower wavelength regions as shown in Fig 9(A)), as multipoles have been created in the plasmonic particle [shown in Fig 9(B) and 9(C)] even in the Rayleigh regime.

In absence of the heterogenous background medium, as depicted in Fig 4(A), the chiral and plasmonic particles (d = 200nm) experience mutual attraction (instead of dipole-dipole repulsion [18]) for the higher wavelength regions as shown in Fig 4(B). In the lower wavelength regions, the repulsion of the chiral-plasmonic heterodimers placed in air has been explained in the previous section considering the impact of nearly trapped chiral particle in the dimer set (which is a bit different in comparison with ref [18, 23]). The challenging part was to obtain repulsive binding force for higher wavelength regions when both the particles are better described as pure Rayleigh particles. This repulsive force in the higher wavelength regions has been achieved only when the particles are put in a heterogeneous background and the light angles have been changed only a little bit (let us consider the 15-degree case). Due to the presence of both the chiral-plasmonic particles in the set-up (the multiple scattering effect) and high non-paraxial angles between the beams and the heterogenous background, multipoles [as shown in Fig 9(B) and 9(C)] have been created in the plasmonic particle (which is counterintuitive for Rayleigh sized particles). The plasmonic multipolar particle is supporting the directed (towards -z direction) linear increase of momentum [55, 56] of the emitted photons.

For our case, the plasmonic particle is experiencing pulling force (towards its nearest unperturbed light source) in the dimer set-up of Fig 8(A) for the combined impacts of three catalysts together: (i) the presence of high non-paraxial angled beams (that creates multipolar response), (ii) transfer of Minkowski momentum (of emitted photon) to the plasmonic particle and (iii) repulsive force in the near field in presence of the chiral particle due to mutual scattering. If any of the aforementioned catalysts is removed, mutual repulsion at higher wavelength region or individual pulling type force vanishes. In summary, the dimer set, illustrated in Fig 8 (B) and 8(C), experiences a repulsive binding force even in higher wavelength regions, as depicted in Fig 8(A), due to the pulling force exerted on each particle. Here, the pulling force refers to the force directing a particle towards the closest unperturbed light source.

Finally, considering the far-field case where the interparticle distance is 700 nm, as depicted in Fig 8(H)–8(J), we observe a similar phenomenon. The broad-band repulsive thrust between the dimers is a result of the individual pulling forces acting on the Rayleigh-sized chiral and

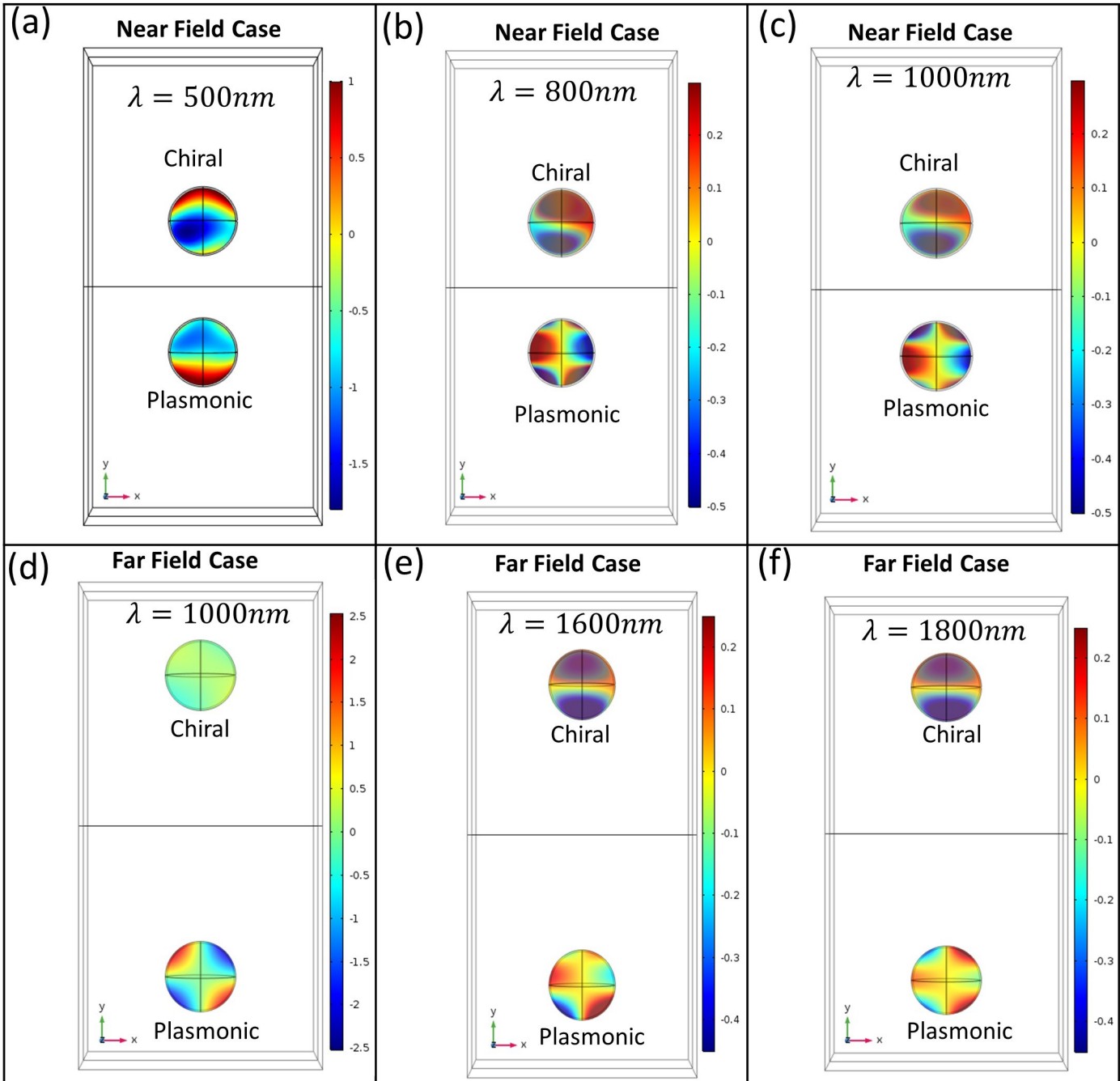

**Fig 9.** (a-c) Electric field profile for setup Fig 8(a) (near field) at wavelength 500*nm*,800*nm*, & 1000nm respectively (where pulling force occurs). (d-e) Electric field profile for setup Fig 8(h) (far field) at wavelength 1000*nm*,1600*nm*, & 1800nm respectively (where pulling force occurs).

plasmonic particles embedded in a heterogeneous background. Our analysis indicates that the physical reasoning for this occurrence aligns closely with the near-field case. The multipolar behavior of the Rayleigh-sized plasmonic particle and the dipolar behavior of the Rayleigh-sized chiral particle are further detailed in Fig 9(D)–9(F). This observation substantiates that increasing the refractive index of the lower medium [water (n = 1.33) replaces air (n = 1) as a

lower medium] can effectively intensify the repulsive thrust for the configurations illustrated in Fig 8(A) and 8(H).

Given the insights and observations discussed, the OBF observed in this study, occurring between chiral-plasmonic particles, is termed as a 'modified' longitudinal OBF instead of the usual well-known longitudinal OBF. This distinction is made in contrast to the conventional longitudinal OBF, typically observed with a single light source or between two simplistic dipolar entities like dielectric dipole-sized objects, in the presence of counter-propagating waves [18, 23].

## 5. Conclusion

In brief, our study in this article presents a novel observation in light-matter interaction: the manipulation of counter-intuitive repulsive force in both near and far-field regions between plasmonic-chiral heterodimers using linearly polarized counter-propagating plane waves. Our exploration involved examining the impact on this force in two different environments—homogeneous and heterogeneous backgrounds. Initially, in a homogeneous (air) environment, applying a single plane wave to our dimer setup did not reverse the OBF. However, the introduction of two oppositely directed linearly polarized plane waves resulted in repulsion between the particles, considered as a 'modified' longitudinal binding force. This observed repulsive thrust of the heterodimers in presence of two counter propagating light beams cannot be solely attributed to photon or particle like nature of light during the light-matter interaction. For this specific heterodimer set-up of chiral-plasmonic, it appears that the wave nature of light is playing a crucial role during the light-matter interaction. Finally, our proposed set-up has been investigated further in heterogenous (air-water interface) background, with slight variations in light angle. Our observation for this interface case is suggesting the transfer of Minkowski momentum of photon (particle type behavior of light) to each optically pulled Rayleigh or dipolar particle of the dimer set, which ultimately leads to a broad-band giant repulsive binding force between the dimers. Notably, in absence of the other particle in the cluster, a single half-immersed particle does not experience the pulling force for the broadband spectrum. The underlying reason of the observed repulsive thrust of the dimers for both the scenarios is attributed to "modified" longitudinal OBF. We believe that controlling the repulsion of two objects (near- and far-field 'modified' longitudinal binding forces of plasmonic-chiral hybrid dimers) by utilizing two simple unstructured light beams and the background medium presents a unique strategy for optical manipulation with an impactful prospect in the studies of nanoparticle interactions, actuators, particle categorization, and coalescence.

## Supporting information

**S1 File.**
(DOCX)

## Author Contributions

**Conceptualization:** Sudipta Biswas, M. R. C. Mahdy, Saikat Chandra Das.

**Formal analysis:** Sudipta Biswas.

**Funding acquisition:** M. R. C. Mahdy.

**Investigation:** Sudipta Biswas, Saikat Chandra Das, Md. Ariful Islam Bhuiyan.

**Methodology:** Sudipta Biswas, M. R. C. Mahdy, Saikat Chandra Das.

**Project administration:** Sudipta Biswas.

**Resources:** Sudipta Biswas, M. R. C. Mahdy.

**Software:** Sudipta Biswas, Saikat Chandra Das, Mohammad Abir Talukder.

**Supervision:** M. R. C. Mahdy.

**Validation:** M. R. C. Mahdy, Mohammad Abir Talukder.

**Visualization:** Sudipta Biswas, Md. Ariful Islam Bhuiyan, Mohammad Abir Talukder.

**Writing – original draft:** Sudipta Biswas, M. R. C. Mahdy, Saikat Chandra Das.

**Writing – review & editing:** M. R. C. Mahdy, Md. Ariful Islam Bhuiyan, Mohammad Abir Talukder.

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
