## [Decision Letter · Decision Letter 0]

20 Aug 2023

PONE-D-23-24071Controlling the Counterintuitive Optical Repulsive Thrust of Nano Dimers with Counter Propagating Type Waves and Background MediumPLOS ONE

Dear Dr. Chowdhury,

Thank you for submitting your manuscript to PLOS ONE. After careful consideration, we feel that it has merit but does not fully meet PLOS ONE’s publication criteria as it currently stands. Therefore, we invite you to submit a revised version of the manuscript that addresses the points raised during the review process.

We look forward to receiving your revised manuscript.

Kind regards,

Yuan-Fong Chou Chau

Academic Editor

PLOS ONE

Journal Requirements:

   "Funding: M.R.C. Mahdy acknowledges the support of internal CTRGC grant 2021-22 and 2022-23 of North South University (Approved by the members of BOT), Bangladesh."

Reviewers' comments:

Reviewer's Responses to Questions

**Comments to the Author**

1. Is the manuscript technically sound, and do the data support the conclusions?

Reviewer #1: Partly

Reviewer #2: Yes

Reviewer #3: Partly

2. Has the statistical analysis been performed appropriately and rigorously? 

Reviewer #1: Yes

Reviewer #2: Yes

Reviewer #3: I Don't Know

3. Have the authors made all data underlying the findings in their manuscript fully available?

Reviewer #1: Yes

Reviewer #2: Yes

Reviewer #3: No

4. Is the manuscript presented in an intelligible fashion and written in standard English?

Reviewer #1: No

Reviewer #2: Yes

Reviewer #3: No

5. Review Comments to the Author

Reviewer #1: This manuscript introduces a new discovery in the realm of light-matter interaction: the manipulation of the counter-intuitive repulsive force within the proximity and distant regions of plasmonic-chiral heterodimers. This is achieved by subjecting them to linearly polarized counter-propagating plane waves. To influence and manage this repulsive force exerted on the dimers, the authors have explored two different contexts, homogeneous and heterogeneous backgrounds in our configurations.

While I find the content intriguing, I am unable to endorse its publication in its current state. The following points outline my concerns and questions:

1. While the concept and content of this manuscript are innovative, the manner in which it is presented makes it challenging to grasp the overarching structure. The document leans heavily towards a descriptive style, yet the logical flow poses difficulties for readers to follow. The authors may consider revising the article's structure to achieve a more organized and coherent presentation.

2. Following 1, it is advisable for the authors to emphasize the underlying physical mechanism or rationale behind this altered longitudinal OBF. Allocating a distinct section to elucidate this aspect could assist readers in concentrating on and comprehending it.

3. The authors should enhance the substantiation of their claim that "By increasing the refractive index of the lower medium, it is indeed possible to further enhance the repulsive thrust for the setups shown in Fig. 8(a) and (h)," as pointed out in line 621.

4. Despite the authors' inclusion of a 15-degree incident angle in both Fig. 2 and Fig. 8, there are discrepancies apparent in the schematic where certain angles do not conform to 15-degree. It is recommended that the authors address and rectify this inconsistency."

5. What is the reason behind having four legends in Fig. 4(b) when only two curves are actually shown?

6. The x-axis unit in Fig. 4(b) is labeled incorrectly.

7. There are some symbols typo such as “k” on line 237 should be “\\kappa” and $\\bar{P}$ in line 416 should be $\\bar{p}$.

8. There are grammatical errors in many sentences, such as "In differentiate, it shows up that the wave nature of light is playing a crucial part for the light-matter interaction of our proposed setup."

9. Enhancement of the resolution quality in Fig. 8 is recommended, as the current presentation lacks clarity in showcasing finer details.

10. The authors should consider revising their use of abbreviations to avoid repetition. For instance, on page 2, the authors already introduce the abbreviation "OBF" for "optical binding force" on line 97. However, on line 226, instead of utilizing the established abbreviation "OBF," they continue to abbreviate "optical binding force" again.

Reviewer #2: This paper provides a method to control the repulsion and attraction behavior of

nanoparticles using a “modified” longitudinal optical binding force. It demonstrates that the

broadband, especially in higher wavelength regions, repulsive binding force can be achieved

using counterpropagating waves in heterogeneous background, due to the linear increase of

momentum of the emitted photons, while it is not achievable in homogeneous background.

The theoretical and numerical analysis show corresponding results which support the authors

claim. The paper is recommended for publication upon satisfactorily answering the following

questions.

1. What is the relation between the optical binding force and the electric polarity and

magnetic rotation? At the moment, the repulsive behavior seem to be related to the

similarity between the induced electric polarity and magnetic rotation in chiral and

plasmonic particles. Is it possible to generate optical binding forces using the set-up in

Figs. 1(a) and (b) using plasmonic-chiral dimers? It is important to show that there is

actually conventional optical binding force here and then controlling the repulsive thrust

can be discussed. The current observation seems to suggest the repulsive behavior comes

from particle-particle interaction instead of light-particle interaction.

2. Although it has mentioned several times in the manuscript, it is not clear to the reviewer

why in homogeneous case wave nature of light plays a vital role, but in heterogeneous

case the particle nature plays a vital role. There lacks discussion which points out explicitly

the feature or behavior that relates to wave and particle nature, respectively, based on

observations.

3. Basically, the authors have not explained why the car-person interaction is analogy to the

light-particle interaction in Fig. 1. When persons move away from each other, they must

push the car back as shown in the figures, but when particles move away, the waves cannot

be pushed back. I don't see this as an appropriate analogy.

4. It is highly suggested that the authors summarize and itemize the “modification” in the

so called “modified” optical binding force.

Other minor points:

1. Fig. 4(b) misses the other two curves although they are listed in the legend. It is suggested

to use different line types or markers to plot the curves such that the overlapped data can

still be seen from the figure.

2. It is suggested to mark the plasmonic and chiral particle in Fig. 5.

3. It is suggested to substitute all the line plots in Fig. 8 with higher resolution figures with

readable values on the axes.

Reviewer #3: The manuscript prepared by SUDIPTA BISWAS et al. reported a repulsive force between plasmonic and chiral particles in both near and far field regions when counter-propagating plane waves are illuminated to the particles. But, in my opinion, the manuscript is not mature, so it is not easy to read and understand. The structure of the text is not well organized and scattered for me. The Introduction is too long. I do not feel the need to use the analog of two cars and two persons. Authors repeat the same things multiple times. More critically as a scientific report, some simulation details and results are not fully mentioned and/or explained in the text, for example, Figure 3b, 3c, and Figure 5. Especially Figure 3b, OBF is repulsive even though a single propagating wave is irradiated on the plasmonic-chiral heterodimer, which seems not consistent with author’s claim in this contribution. The discussion regarding to the wave and particle natures of light for the counterintuitive repulsive force (more specifically about so-called three catalysts) is not well examined in the manuscript. Thus, I do not recommend the manuscript to be published in the PLOS One. I strongly suggest the authors consider resubmitting after a basic rearrangement of the manuscript and get proofreading.

6. PLOS authors have the option to publish the peer review history of their article (what does this mean?). If published, this will include your full peer review and any attached files.

Reviewer #1: No

Reviewer #2: No

Reviewer #3: No

---

## [Author Response · Author response to Decision Letter 0]

22 Oct 2023

Response Letter

Dear Editor,

We are submitting the revised version of the manuscript ["Controlling the Counterintuitive Optical Repulsive Thrust of Nano Dimers with Counter Propagating Type Waves and Background Medium" (ID: PONE-D-23-24071)] along with the response letter. We have carefully addressed both the reviewers’ comments and made the consequent revisions and improvements of the manuscript. All changes to our main manuscript article have been marked in red color. We believe that: you will really consider the merit of the proposed results and the current version of the overall work.

We’d like to manifest our sincere thanks to the reviewers for their constructive and informative comments to improve our manuscript, all of which are valuable for the improvement of our work. In the seperate response letter, we list all the comments and our replies one by one.

Sincerely yours,

Dr. Mahdy Rahman Chowdhury.

Associate Professor, North South University.

Ph.D., Optical force and manipulation, National University of Singapore.

- On behalf of all the authors.

---

## [Decision Letter · Decision Letter 1]

28 Nov 2023

Controlling the Counterintuitive Optical Repulsive Thrust of Nano Dimers with Counter Propagating Type Waves and Background Medium

PONE-D-23-24071R1

Dear Dr. Chowdhury,

We’re pleased to inform you that your manuscript has been judged scientifically suitable for publication and will be formally accepted for publication once it meets all outstanding technical requirements.

Kind regards,

Yuan-Fong Chou Chau

Academic Editor

PLOS ONE

Additional Editor Comments (optional):

The paper is well revised and may be accepted for publication.

Reviewers' comments:

Reviewer's Responses to Questions

**Comments to the Author**

1. If the authors have adequately addressed your comments raised in a previous round of review and you feel that this manuscript is now acceptable for publication, you may indicate that here to bypass the “Comments to the Author” section, enter your conflict of interest statement in the “Confidential to Editor” section, and submit your "Accept" recommendation.

Reviewer #1: All comments have been addressed

2. Is the manuscript technically sound, and do the data support the conclusions?

Reviewer #1: Yes

3. Has the statistical analysis been performed appropriately and rigorously? 

Reviewer #1: Yes

4. Have the authors made all data underlying the findings in their manuscript fully available?

Reviewer #1: Yes

5. Is the manuscript presented in an intelligible fashion and written in standard English?

Reviewer #1: Yes

6. Review Comments to the Author

Reviewer #1: (No Response)

7. PLOS authors have the option to publish the peer review history of their article (what does this mean?). If published, this will include your full peer review and any attached files.

Reviewer #1: No

---

## [Editor Report · Acceptance letter]

12 Dec 2023

PONE-D-23-24071R1 

PLOS ONE

Dear Dr. Mahdy, 

I'm pleased to inform you that your manuscript has been deemed suitable for publication in PLOS ONE. Congratulations! Your manuscript is now being handed over to our production team.

Kind regards, 

on behalf of

Dr. Yuan-Fong Chou Chau 

Academic Editor

PLOS ONE